# Graph Coloring via Neural Networks for Haplotype Assembly and Viral Quasispecies Reconstruction

Hansheng Xue,[1] Vaibhav Rajan,[2] and Yu Lin[1]*

[1]School of Computing, Australian National University, Canberra, Australia
[2]School of Computing, National University of Singapore, Singapore
{hansheng.xue,yu.lin}@anu.edu.au, vaibhav.rajan@nus.edu.sg

## Abstract

Understanding genetic variation, e.g., through mutations, in organisms is crucial to unravel their effects on the environment and human health. A fundamental characterization can be obtained by solving the haplotype assembly problem, which yields the variation across multiple copies of chromosomes. Variations among fast evolving viruses that lead to different strains (called quasispecies) are also deciphered with similar approaches. In both these cases, high-throughput sequencing technologies that provide oversampled mixtures of large noisy fragments (reads) of genomes, are used to infer constituent components (haplotypes or quasispecies). The problem is harder for polyploid species where there are more than two copies of chromosomes. State-of-the-art neural approaches to solve this NP-hard problem do not adequately model relations among the reads that are important for deconvolving the input signal. We address this problem by developing a new method, called `NeurHap`, that combines graph representation learning with combinatorial optimization. Our experiments demonstrate substantially better performance of `NeurHap` in real and synthetic datasets compared to competing approaches.

## 1 Introduction

Our genetic material is organized as sequences of DNA or RNA molecules (nucleotides) which form three-dimensional structures (chromosomes) within our cells. Most organisms have multiple highly similar copies of chromosomes in their cells (e.g., humans have 2). Variations in genetic sequences lead to the emergence of new species during evolution and are also known to be associated with many diseases (e.g., cancer). There are many possible ways in which such variations can occur; the simplest among them is a *mutation* or a change in the nucleotide at a specific location in the DNA or RNA sequence. A Single Nucleotide Polymorphism (SNP) refers to a mutation in at least one of the copies which renders the copies nonidentical at that point. An ordered list of SNPs on a single chromosome is called a haplotype [Schwartz, 2010]. Haplotypes provide a signature of genetic variability and thus inform us about disease susceptibilities and evolutionary patterns (e.g., of viruses). These studies in turn pave the way for personalized medicine and effective drug development against viruses.

The problem of inferring haplotypes from high-throughput sequencing data is called haplotype assembly or phasing, and is done in multiple stages (see Figure 1). Sequencing data yields multiple copies of short fragments of the entire genomic sequence (called reads, Figure 1b). These reads are noisy due to sequencing errors and their short lengths may span across limited number of SNPs. This makes the problem of haplotype phasing challenging. The reads are first aligned to a reference genome. This step indicates positions that are different across reads and thus infers the potential locations of SNPs. All other positions are discarded to obtain the SNP matrix (Figure 1c). This

---

*Corresponding author.

36th Conference on Neural Information Processing Systems (NeurIPS 2022).

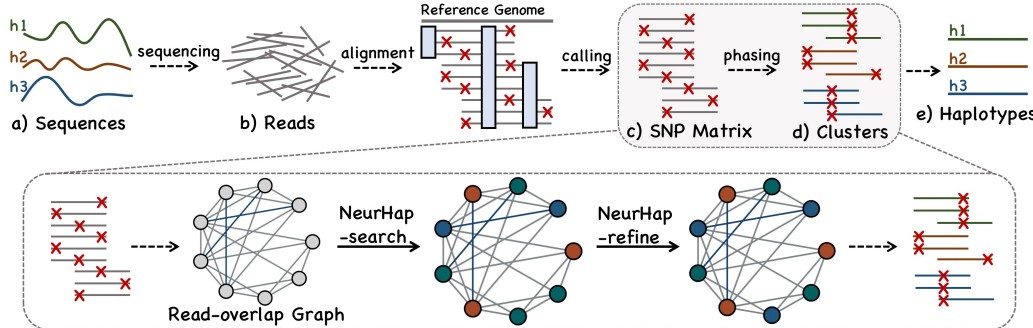

Figure 1: The pipeline of reference-based polyploid haplotypes reconstruction and `NeuralHap`. Haplotype phasing is formulated as a graph coloring problem by constructing the Read-overlap graph. `NeurHap` consists of `NeurHap`-search, a graph neural network to learn vertex representations and color assignments, and `NeurHap`-refine, a local refinement strategy to further adjust colors.

matrix may be viewed as an oversampled mixture of noisy reads (restricted to SNPs). Each mixture component represents a single haplotype and thus should have SNPs at the same locations.

In diploid species, containing two copies of chromosomes, there are two haplotypes to be inferred. This problem has been studied extensively [Browning and Browning, 2011]. In polyploid species, containing more than two copies of chromosomes (and thus more than two haplotypes), the problem is more challenging due to dramatic increase in search space [Van de Peer et al., 2017, Abou Saada et al., 2022, Jablonski and Beerenwinkel, 2021]. In reconstruction of virus strains, called viral quasispecies, from viral populations, similar challenges arise. Moreover, unknown population sizes and imbalanced abundances pose additional difficulties [Jablonski and Beerenwinkel, 2021].

Existing approaches for haplotype phasing of polyploid species and viral quasispecies often group reads in the SNP matrix into clusters that correspond to different haplotypes, respectively. In an ideal case, all reads from the same cluster should be consistent with respect to SNPs as they all belong to the same haplotype. In reality inconsistencies occur due to sequencing errors in reads. Therefore, a minimum error correction (MEC) score [Lippert et al., 2002b] is used to measure the discrepancy between the consensus haplotypes and their associated reads within each cluster (see Figure 1e). It is NP-hard to optimize the MEC score [Zhang et al., 2006], and a number of combinatorial optimization heuristics have been proposed to approximate the optimal MEC score [Zhang et al., 2020].

More recently, the first neural network-based learning framework, named GAEseq [Ke and Vikalo, 2020b], was proposed to phase haplotypes for polyploid species and viral quasispecies. CAECseq was later developed using a convolutional auto-encoder which captures spatial relationships between SNPs and enables clustering reads obtained from highly similar genomic regions [Ke and Vikalo, 2020a]. Both GAEseq and CAECseq showed improved results compared to previous approaches. A major limitation of both CAECseq and GAEseq is that they cannot capture implicit relations among different reads. These methods have two independent steps (embedding and clustering) which makes the haplotype phasing results unstable. Besides, sparsity of the SNP matrix makes haplotype phasing for polyploids more challenging for these methods as well.

In this paper, we propose an approach based on graph representation learning for haplotype phasing of polyploid species and viral quasispecies. We formulate the haplotype phasing problem as a graph coloring problem, where the colors indicate haplotypes. The graph is constructed from the SNP matrix where vertices are reads and two edge types are defined based on pairwise consistency and conflicts with respect to SNPs in the reads. Message passing-based neural networks are trained to minimize a loss designed to obtain a color assignment that maximizes consistent edges and minimizes conflicting edges. The network learns vertex representations and through them, an initial color assignment. A local refinement strategy is then applied to adjust node colors in order to minimize MEC scores. Thus, in contrast to previous neural approaches that first learn representations and then cluster, our approach models the problem requirements in all steps. As a result, our model achieves better MEC scores, is more stable and also performs well on the challenging cases of polyploids and viral quasispecies. In summary, our contributions are:

- We provide a unique formulation of the haplotype phasing problem as a graph coloring problem, and develop an algorithm based on graph representation learning and combinatorial optimization.
- Our approach consists of `NeurHap`-search, a graph neural network to learn vertex representations and color assignments followed by `NeurHap`-refine, a local refinement strategy to adjust colors and optimize MEC scores.
- Extensive experiments on synthetic and real datasets demonstrate that our new method `NeuralHap` significantly outperforms state-of-the-art phasing methods for both polyploid species and viral quasispecies.

## 2 Related Work

**Haplotype Phasing.** The aim of haplotype phasing of polyploid species and viral quasispecies is to group reads into homogeneous clusters that corresponds to different haplotypes, respectively. The minimum error correction (MEC) score [Lippert et al., 2002b] is introduced to measure the total discrepancy of reads in all clusters but is NP-hard to be optimised [Zhang et al., 2006]. Haplotype phasing for diploid species (i.e., reconstructing two haplotypes) has been extensively studied in the last two decades and a number of combinatorial optimization heuristics have been proposed to approximate the optimal MEC score, such as BNB [Wang et al., 2005], HapCUT [Bansal and Bafna, 2008], HASH [Bansal et al., 2008], RefHap [Duitama et al., 2012] ProbHap [Kuleshov, 2014], HapCUT2 [Edge et al., 2017] and others, and refer to [Zhang et al., 2020] for a recent review on phasing diploid species.

Haplotype phasing for polyploid species (i.e., reconstructing more than two haplotypes) becomes more computationally challenging as it requires a much larger search space compared to phasing two haplotypes for diploid species. A limited number of phasing methods work for polyploid species, e.g., HapCompass [Aguiar and Istrail, 2012], SDhaP [Das and Vikalo, 2015], H-PoP [Xie et al., 2016], AltHap [Hashemi et al., 2018], refer to [Abou Saada et al., 2022] for a recent review. Haplotype phasing for viral quasispecies is very similar to the problem of phasing polyploid species. While haplotypes in polyploid species typically have uniform abundances, the different haplotypes (strains) in viral quasispecies may have varying abundances. Quite a few tools have also been proposed for haplotype phasing of viral quasispecies, such as ViSpA [Astrovskaya et al., 2011], ShoRAH [Zagordi et al., 2011], QuRe [Prosperi and Salemi, 2012], QuasiRecomb[Töpfer et al., 2013], PredictHaplo [Prabhakaran et al., 2014], aBayesQR [Ahn and Vikalo, 2018], TenSQR [Ahn et al., 2018], refer to [Jablonski and Beerenwinkel, 2021] for a recent review.

More recently, deep learning models have been introduced into haplotype phasing for polyploid species and viral quasispecies. GAEseq [Ke and Vikalo, 2020b] uses a graph auto-encoder model on the constructed reads-SNPs bipartite network to model the relations between reads and SNPs. CAECseq [Ke and Vikalo, 2020a] uses a convolutional auto-encoder model to represent reads as low-dimensional features and then employs a clustering algorithm to group these reads. Note that GAEseq and CAECseq can be directly used to phase haplotypes for both polyploid species and viral quasispecies. Experimental results on both simulated and real datasets showed the superior results of GAEseq and CAECseq (in terms of MEC scores) compared to previous approaches for haplotype phasing for both polyploid species and viral quasispecies [Ke and Vikalo, 2020b,a]. However, implicit relations among different reads have not been fully captured by GAEseq and CAECseq, especially when embedding and clustering are modelled separately and the SNP matrix is sparse.

**Neural Networks on Graphs.** Most existing graph neural networks can be explained as a message-passing based graph learning model which recursively combines learned features/messages from their neighbors [Cui et al., 2019, Cai et al., 2018, Gilmer et al., 2017]. Popular methods include GCN [Kipf and Welling, 2017], GraphSAGE [Hamilton et al., 2017], GAT [Veličković et al., 2018] and GIN [Xu et al., 2019]. All these methods make the homophily assumption that similar nodes in the graph should be embedded close together. However, graph coloring aims to assign pairwise nodes with distinct colors for each edge of the graph, which is opposite to the homophily assumption. An intuitive way to integrate GNN models into the graph coloring challenge is to adjust the loss function, such as GNN-GCP [Lemos et al., 2019], RUN-CSP [Toenshoff et al., 2019], and PI-GNN [Schuetz et al., 2022]. However, existing GNN-based graph coloring models cannot be implemented to the read-overlap graph directly because they cannot handle conflicting and consistent edges simultaneously.

## 3 Methodology

**Overview.** In this paper, we propose the model of neural networks for graph coloring optimization to solve the haplotype reconstruction problem, called `NeurHap`. `NeurHap` mainly contains three steps, i) constructing the *read-overlap graph*; ii) global coloring searching via iterative neural networks model, `NeurHap-search`; iii) local refinement to fine tune final coloring, `NeurHap-refine`.

**Notations.** Let $k$ be the number of haplotypes in a cell of polyploid species (aka. *ploidy*) or the number of strains in viral quasispecies. For example, human and other mammals contain two haplotypes (diploid, $k=2$); plants have more than 2 haplotypes (e.g., California redwood has six copies of each chromosome, hexaploid, $k=6$). Viral quasispecies mixing 5 distinct HIV strains will have $k = 5$.

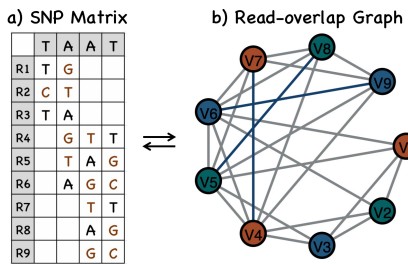

Single nucleotide polymorphisms (SNPs) refer to positions where not all haplotypes have the same alleles. Given the alignment of reads to a reference genome, the SNP columns can be identified by removing the columns with identical alleles. The remaining alignment is referred to as a $m \times n$ SNP matrix $\mathcal{R}$ where $m$ denotes the number of reads and $n$ is the number of SNPs. The haplotype reconstruction aims to group $m$ reads into $k$ clusters,

Figure 2: A toy example of constructing *read-overlap graph* with conflict edges (in grey) and consistent edge (in blue).

$\{C_1, C_2, \ldots, C_k\}$, that correspond to $k$ haplotypes, $\{\mathcal{H}_1, \mathcal{H}_2, \ldots, \mathcal{H}_k\}$, respectively. Once reads are grouped into clusters, the haplotype $\mathcal{H}_i$ can be reconstructed from reads in $C_i$ using a simple consensus voting. In the ideal case, all the reads from the cluster $C_i$ will be all consistent with $\mathcal{H}_i$. In reality, this is not the case and thus a minimum error correction (MEC) score [Lippert et al., 2002b] is introduced to measure the discrepancy between the reads in the cluster $C_i$ and the consensus haplotypes $\mathcal{H}_i$ in all clusters. Given the grouping of reads into $k$ clusters $\{C_1, C_2, \ldots, C_k\}$, the corresponding MEC score can be computed as

$$\text{MEC}(C_1, C_2, \ldots, C_k) = \sum_{i=1}^{k} \sum_{R_j \in C_i} HD(\mathcal{H}_i, R_j) \tag{1}$$

where $HD(\cdot)$ is the Hamming distance function. Note that $HD(\mathcal{H}_i, R_j)$ for $R_j \in C_i$ can only be computed when we know all the reads in the cluster $C_i$ and use them to derive the consensus haplotype $\mathcal{H}_i$ of $C_i$. The main challenge in haplotype phasing is to find the grouping of reads into $\{C_1, C_2, \ldots, C_k\}$ such that the MEC score is minimized.

Two reads are called *overlapping* if they span over common SNP positions otherwise *non-overlapping*. Given any two reads, the relationship between them belongs to one of three cases, *consistent*, *conflict*, or *ambiguous*. While the relationship between two non-overlapping reads is always *ambiguous*, we further introduce two parameters $p$ and $q$ to define the relationship between two overlapping reads to account for sequencing errors and alignment ambiguity. Two overlapping reads are *consistent* if they overlap at least $p$ positions and have the same alleles over all overlapping positions; are in *conflict* if they differ on at least $q$ overlapping positions; and are *ambiguous* otherwise. The term '*ambiguous*' means that there is not enough evidence to support that these two reads should belong to the same haplotype ('*consistent*') or should belong to the different haplotypes ('*conflict*'). For example, in Figure 2, $R_4$ and $R_7$ are *consistent*, $R_1$ and $R_2$ are in *conflict*, and $R_1$ and $R_4$ are *ambiguous*. In an ideal case, all the overlapping reads in the same cluster must be *consistent*, i.e., if two reads are in *conflict*, they must belong to different clusters. This observation naturally motivates us to build a read-overlap graph to model all reads as vertices and the important pairwise relationships (i.e., *consistent* and *conflict*) between overlapping reads as edges. Moreover, if we use $k$ colors to represent the $k$ clusters of reads, the problem of haplotype phasing is reduced to a graph coloring problem on the read-overlap graph. For example, in Figure 2, the minimum MEC is achieved by grouping nine reads into three clusters, $C_1 = \{R_1, R_4, R_7\}$, $C2 = \{R_2, R_5, R_8\}$ and $C_3 = \{R_3, R_6, R_9\}$, which correspond to three distinct colors on corresponding vertices in the read-overlap graph, respectively.

## 3.1 Graph Coloring over the Read-overlap Graph.

The *read-overlap graph* $\mathcal{G} = (\mathcal{V}, \mathcal{E}_=, \mathcal{E}_{\neq})$ is constructed in this step. Here, the vertex set $\mathcal{V}$ denotes all reads, the edge set $\mathcal{E}_=$ represents all pairwise consistent relationships between overlapping reads, and the edge set $\mathcal{E}_{\neq}$ refers to the pairwise conflict relationships between overlapping reads.

Now we are ready to reduce the problem of haplotype phasing to the graph coloring problem on the read-overlap graph. Recall that the haplotype phasing problem aims to group reads into $k$ clusters such that reads from the same cluster are as consistent as possible. If we employ $c(v)$ to represent one out of $k$ colors assigned to a read $v$ (i.e., one of the $k$ clusters that $v$ belongs to), two reads $R_i$ and $R_j$ are in the same cluster if and only if two corresponding vertices $v_i$ and $v_j$ have the same color, i.e., $c(v_i) = c(v_j)$. Now the graph coloring problem needs to assign a color to $c(v)$ for every vertex $v \in \mathcal{V}$ to minimise the MEC score under the constraints that any two conflicting reads have two different colors and any two consistent reads have the same color.

$$\min \ \text{MEC}(c(v_1), c(v_2), \ldots, c(v_n)) = \min \sum_{i=1}^{k} \sum_{c(v_j)=i} \text{HD}(\mathcal{H}_i, R_j)$$

$$\text{s.t.,} \quad \begin{cases} \forall (v_i, v_j) \in \mathcal{E}_{\neq}, c(v_i) \neq c(v_j) \\ \forall (v_i, v_j) \in \mathcal{E}_=, c(v_i) = c(v_j) \end{cases} \tag{2}$$

Note that the above graph coloring problem is different from the classical graph coloring problem [Pardalos et al., 1998] in combinatorial optimization. While all the edges are conflicting edges in the classical graph coloring problem, the above problem formulation in the equation 2 has constraints for both conflicting and consistent edges. In the following section, we will show how to model these constraints using neural networks.

## 3.2 Network-based Global Search and Combinatorial Optimization-based Local Refinement

**Satisfying Constraints.** As the vertices of the read-overlap graph need to be colored to satisfy the constraints in the equation 2, we further reduce the graph coloring problem to a constraints satisfaction problem inspired by RUN-CSP [Toenshoff et al., 2019]. Graph neural networks (GNNs) are designed to follow homophily constraints such that similar vertices in the graph are embedded close to each other (i.e., same colors). While this is true for consistent constraints in the read-overlap graph, the conflicting constraints impose heterozygous constraints, i.e., vertices connected by a conflicting edge should have very different embeddings (i.e., different colors). Therefore, given $k$ distinct colors, we introduce two 0/1 metrics in $\mathbb{R}^{k \times k}$ for incorporating the above two different coloring constraints, $\mathcal{M}_{\neq}$ (for conflicting constraints) and $\mathcal{M}_=$ (for consistent constraints). The conflict-constraint matrix $\mathcal{M}_{\neq}$ denotes the binary conflict relationships among $k$ colors, i.e., $\mathcal{M}_{\neq}(i, j) = 1$ if $i \neq j$ and 0 otherwise, for any $i, j \in \{1, ..., k\}$. The consistent-constraint matrix $\mathcal{M}_=$ denotes the consistent relationships between $k$ colors, i.e., $\mathcal{M}_=(i, j) = 1$ if $i = j$ and 0 otherwise, for any $i, j \in \{1, ..., k\}$.

Assume that the coloring-assignment matrix $P \in \mathbb{R}^{|\mathcal{V}| \times k}$ is a matrix that represents the coloring assignment probability (over $k$ colors) for each vertex $v$ in its corresponding row $P(v)$. For any conflicting edge $(v_i, v_j) \in \mathcal{E}_{\neq}$ in the read-overlap graph, we aim to assign different colors to $v_i$ and $v_j$ and thus maximize the sum of joint-probabilities with different colors, i.e., $P(v_i) \mathcal{M}_{\neq} P(v_j)^{\mathsf{T}}$. Symmetrically, for any consistent edge $(v_i, v_j) \in \mathcal{E}_=$ in the read-overlap graph, we aim to assign the same color to $v_i$ and $v_j$ and thus maximize the sum of joint-probabilities with same colors, i.e., $P(v_i) \mathcal{M}_= P(v_j)^{\mathsf{T}}$. In summary, the unsupervised objective function can be formulated as follows:

$$\mathcal{L} = -\frac{1}{|\mathcal{E}_{\neq}|} \sum_{(v_i, v_j) \in \mathcal{E}_{\neq}} \log(P(v_i) \mathcal{M}_{\neq} P(v_j)^{\mathsf{T}}) - \lambda \cdot \frac{1}{|\mathcal{E}_=|} \sum_{(v_i, v_j) \in \mathcal{E}_=} \log(P(v_i) \mathcal{M}_= P(v_j)^{\mathsf{T}}). \tag{3}$$

Here, $\lambda$ controls the importance of consistent constraints compared to conflicting constraints. We will now show how to use representation learnings to derive a coloring-assignment matrix $P$ that optimizes the above objective function.

**Global Search:** `NeurHap`**-search.** To derive a coloring-assignment matrix $P$, we propose to use an iterative message passing-based representation learning model to capture the structural information of the read-overlap graph. The message-passing learning model mainly contains three operation functions, message-learning, aggregation, and combine operator. Given trainable $d$-dimensional

embeddings for every nodes $\{h(v_1), h(v_2), ..., h(v_n)\}, h(v_i) \in \mathbb{R}^d, \ v_i \in \mathcal{V}$, which are initialized by randomly sampling from a uniform distribution, the message passing model can be formulated as:

$$\begin{aligned}
h(v_i) &= \text{COMBINE}(m(v_i), h(v_i)), \\
m(v_i) &= \text{AGGREGATE}(\{\text{MESSAGE}(h(v_i), h(v_j)) : v_j \in N(v_i)\}).
\end{aligned} \tag{4}$$

Here, $m(v_i)$ is the learned messages from the neighbors of $v_i$, and $N(v_i)$ is the neighbors of $v_i$ with respect to conflict edges. COMBINE$(\cdot)$ is a combine function and AGGREGATE$(\cdot)$ denotes an aggregation function. To search for a simple model, we adopt a recent message updater and mean operator $(m(v_i) = \frac{1}{|N(v_i)|} \sum_{v_j \in N(v_i)} h(v_j))$ as combine and aggregate functions respectively. MESSAGE$(\cdot)$ represents the learnable message function, e.g. MESSAGE$(h(v_i), h(v_j)) =$ MLP$(h(v_i)||h(v_j))$. Two linear layers with activation function (e.g., ReLU) are selected to construct the MLP layer. A simple linear decoder can be used to map the learned node embeddings to the probability of colors: $P(v_i) = \text{DEC}(h(v_i))$. The message-passing model is iteratively trained for $t$ times in each epoch to generate reliable features for each node. The pseudocode for the global search process of `NeurHap` is as follows:

---

**Algorithm 1:** The Global Search Algorithm `NeurHap`-search

**Data:** SNP matrix $\mathcal{R}$; number of iteration $t$; number of polyploids $k$; dimension of hidden features $d$.
**Result:** Assignments $\mathcal{Y}$.
1   $\mathcal{E}_{\neq}, \mathcal{E}_{=} \leftarrow$ Equation 1     // Construct conflict and consistent edge set
2   $\mathcal{M}_{\neq}(k), \mathcal{M}_{=}(k)$     // Initialize coloring constraints
3   $h \leftarrow \mathbb{R}^d \sim [0, 1)$     // Initialize by a uniform distribution
4   **for** $e$ epochs **do**
5     **for** $t$ iterations **do**
6       $\overline{m}(v_j) = \text{msg}(h(v_i), h(v_j)) = \text{MLP}(h(v_i)||h(v_j))$     // Compute message from $h(v_i)$ and $h(v_j)$
7       $m(v_i) = \text{agg}(\overline{m}(v_j) : v_j \in N(v_i))$    // Aggregate messages from neighbors of $v_i$
8       $h(v_i) = \text{comb}(m(v_i), h(v_i))$ // Combine messages from $h(v_i)$ and $m(v_i)$
9     **end**
10    $P \leftarrow \text{dec}(h(v_i))$     // Compute coloring assignment probs
11    $\mathcal{L} \leftarrow$ Equation 3     // Compute conflict loss
12    $\mathcal{Y} \leftarrow P$     // Compute coloring assignment
13 **end**

---

After optimizing the objective function with `NeurHap`, we can obtain an initial coloring assignment for vertices that satisfy the constraints in the equation 2 in the read-overlap graph. However, the objective function in equation 2 (i.e., the MEC score) may not be optimized as there may exist multiple coloring assignments that satisfy all constraints. Therefore, we run an additional local refinement step to further optimise the objective function in equation 2.

**Local refinement: `NeurHap`-refine.** This step mainly searches for possible color adjustments of individual vertices given their associated conflicting and consistent constraints. More specifically, if an individual vertex can be assigned a color different from its current color without violating any of the associated conflicting constraints with the neighboring vertices, the color is changed if a better MEC score is obtained by the change. The refinement algorithm, `NeurHap`-refine, iteratively explores these possible color adjustments of individual vertices. Refer to Appendix A.1 for the pseudocode.

## 4   Experiments

**Dataset.** To evaluate the proposed method `NeurHap`, we compare `NeurHap` with state-of-the-art baselines for both polyploid species and viral quasispecies. i) *Polyploid species*: The Solanum Tuberosum is Tetraploid (*k=4*) and the datasets of Solanum Tuberosum contains both simulated dataset **Sim-Potato** and real-world dataset **Real-Potato**, both downloaded from [Ke and Vikalo, 2020a,b]. **Sim-Potato** contains 40 sub-datasets, which contains ten different samples sequenced at four distinct coverages (5X, 10X, 20X, and 30X). **Real-Potato** is the Chromosome *5* capture-seq

data of a small solanum tuberosum population available at NCBI (accession SRR6173308 [2]). Ten samples are generated by randomly selecting ten genomic regions as the reference genome. ii) *Viral Quasispecies*: Three viral quasispecies datasets are downloaded from SAVAGE[3] [Baaijens et al., 2017], including the human immunodeficiency virus (**5-strain HIV**, *k=5*), the hepatitis C virus (**10-strain HCV**, *k=10*), and the zika virus (**15-strain ZIKV**, *k=15*). Ten samples are generated by randomly sampling from each of these three datasets. In this paper, we use BWA-MEM [Li, 2013] to align reads to the reference genome and use the same tool described in CAECseq and GAEseq [Ke and Vikalo, 2020a,b] to derive the SNP matrix from the above alignment to ensure a fair comparison.

**Baseline algorithms.** GAEseq [Ke and Vikalo, 2020b] and CAECseq [Ke and Vikalo, 2020a] are two state-of-the-art approaches that work on both haplotype assembly and viral quasispecies reconstruction. We included two additional methods, H-PoP [Xie et al., 2016], AltHap [Hashemi et al., 2018], that specifically work on haplotype assembly for polyploid species. We also included one additional method, TenSQR [Ahn et al., 2018], that specifically works on viral quasispecies reconstruction. Many other specific methods are not included in this study because GAEseq [Ke and Vikalo, 2020b] and CAECseq [Ke and Vikalo, 2020a] have recently demonstrated their superior performance against other baselines in both haplotype assembly and viral quasispecies reconstruction.

**Experimental setup.** The minimum error correction (MEC) score, given in equation 1, is adopted as the evaluation metric [Lippert et al., 2002a] for both haplotype assembly and viral quasispecies reconstruction. Following the experimental setup in [Ke and Vikalo, 2020a,b], all the algorithms run ten times on each input dataset and the lowest MEC score is reported. The initial number of polyploids $k$ is known: $k = 4$ for both Sim-Potato and Real-Potato, $k = 5$ for 5-strain HIV, $k = 10$ for 10-strain HCV, and $k = 15$ for 15-strain ZIKV. The default settings of `NeurHap` hyperparameter are as follows. The representation dimensions are all empirically set to be 32. The number of iteration $t$ in `NeurHap`-search is set to be 10 as default. The parameter $\lambda$ chooses 0.01 as the default value. The default values for parameters $p$ and $q$ are 3 and 5, respectively. The NeurHap model is freely available at `https://github.com/xuehansheng/NeurHap`.

## 4.1 Performance on Polyploid Species data

Table 1 and 2 show that `NeurHap` significantly outperforms state-of-the-art baselines, which achieving the lowest MEC scores on both the Sim-Potato and Real-Potato datasets. For Cov-5X of Sim-Potato, the MEC score obtained by `NeurHap` is 29.9 which is about 3x lower than the lowest scores achieved by baselines (96.2 for CAECseq).

Table 1: Performance comparison on Sim-Potato data.

| Model | #Cov 5X | #Cov 10X | #Cov 20X | #Cov 30X |
|---|---|---|---|---|
| H-PoP | $429.0\pm_{64.1}$ | $933.9\pm_{103.6}$ | $1782.2\pm_{161.8}$ | $2826.9\pm_{180.7}$ |
| AltHap | $610.9\pm_{259.3}$ | $722.3\pm_{179.1}$ | $649.3\pm_{369.4}$ | $1148.2\pm_{509.9}$ |
| GAEseq | $153.7\pm_{20.3}$ | $261.6\pm_{58.7}$ | $372.8\pm_{74.5}$ | $496.9\pm_{128.7}$ |
| CAECseq | $\underline{96.2\pm_{26.9}}$ | $\underline{141.4\pm_{40.7}}$ | $\underline{254.2\pm_{99.7}}$ | $\underline{372.9\pm_{148.9}}$ |
| NeurHap | $\mathbf{29.9\pm_{5.7}}$ | $\mathbf{51.9\pm_{8.2}}$ | $\mathbf{92.6\pm_{10.6}}$ | $\mathbf{142.0\pm_{23.6}}$ |

CAECseq). For Real-Potato, `NeurHap` also achieves the lowest MEC scores on all samples. The average MEC score achieved by `NeurHap` is 371.6 which is significantly lower than the second lowest MEC score obtained by CAECseq, 400. The gap between `NeurHap` and baselines demonstrates the superiority of our model in polyploid haplotype phasing.

Table 2: Performance comparison on Real-Potato data.

| Sample | #1 | #2 | #3 | #4 | #5 | #6 | #7 | #8 | #9 | #10 | Avg. |
|---|---|---|---|---|---|---|---|---|---|---|---|
| Reads | 240 | 389 | 274 | 115 | 141 | 398 | 295 | 284 | 489 | 449 | - |
| SNPs | 294 | 238 | 83 | 23 | 176 | 198 | 456 | 424 | 236 | 410 | - |
| H-PoP | 705 | 525 | 132 | 4 | 240 | 982 | 981 | 766 | 793 | 1413 | $654.1\pm_{435.6}$ |
| AltHap | 746 | 572 | 192 | 9 | 299 | 1295 | 1021 | 982 | 811 | 1311 | $723.8\pm_{451.1}$ |
| GAEseq | 231 | 406 | $\underline{97}$ | $\underline{2}$ | 180 | 873 | 558 | 441 | $\underline{592}$ | 712 | $409.2\pm_{266.6}$ |
| CAECseq | $\underline{229}$ | $\underline{393}$ | 103 | **1** | 172 | $\underline{859}$ | $\underline{522}$ | $\underline{430}$ | 593 | $\underline{698}$ | $\underline{400.0\pm_{260.9}}$ |
| NeurHap | **178** | **343** | **93** | **1** | **163** | **857** | **499** | **384** | **561** | **632** | $\mathbf{371.6\pm_{268.9}}$ |

---

[2]https://www.ncbi.nlm.nih.gov/sra/SRR6173308

[3]https://bitbucket.org/jbaaijens/savage-benchmarks

## 4.2 Performance on Viral Quasispecies data

Table 3 shows the results obtained by `NeurHap` and baselines for reconstructing viral quasispecies on three datasets respectively, 5-strain HIV, 10-strain HCV, and 15-strain ZIKV. In Table 3, `NeurHap` significantly outperforms baselines on all 10 samples in these datasets. `NeurHap` achieves the lowest MEC score in 5-strain HIV (1371.4), which is about 160 lower than the MEC score obtained by CAECseq (1638.5). For 10-strain HCV data, `NeurHap` also achieves the lowest MEC score 1008.1 and the second lowest MEC score is 1144.3 obtained by TenSQR. With increase in the number of haplotypes (strains), performance of CAECseq and GAEseq deteriorates and while that of `NeurHap` improves. `NeurHap` significantly outperforms CAECseq and GAEseq on polyploid haplotypes.

Table 3: Performance comparison on three viral quasispecies datasets.

| Dataset | | #1 | #2 | #3 | #4 | #5 | #6 | #7 | #8 | #9 | #10 | Avg. |
|---|---|---|---|---|---|---|---|---|---|---|---|---|
| | Reads | 967 | 961 | 951 | 961 | 966 | 969 | 962 | 965 | 955 | 971 | - |
| | SNPs | 1617 | 1685 | 1595 | 1605 | 1615 | 1660 | 1619 | 1622 | 1580 | 1653 | - |
| 5-strain HIV | TenSQR | 1920 | 2324 | 1867 | 1896 | 2055 | 1793 | 2125 | 1754 | 1679 | 1757 | $1917.0\pm198.6$ |
| | GAEseq | 1981 | 1953 | 1678 | 1806 | 1905 | 2007 | 1819 | 1746 | 1702 | 1747 | $1834.4\pm119.6$ |
| | CAECseq | 1729 | 1750 | 1787 | 1552 | 1730 | 1622 | 1611 | 1529 | 1519 | 1556 | $1638.5\pm101.5$ |
| | NeurHap | **1307** | **1525** | **1385** | **1265** | **1410** | **1382** | **1393** | **1323** | **1274** | **1450** | $\mathbf{1371.4}\pm\mathbf{81.2}$ |
| | Reads | 500 | 498 | 500 | 499 | 498 | 500 | 499 | 500 | 500 | 500 | - |
| | SNPs | 1770 | 1712 | 1794 | 1749 | 1741 | 1759 | 1786 | 1765 | 1743 | 1808 | - |
| 10-strain HCV | TenSQR | 1081 | 1037 | 1106 | 960 | 1115 | 1015 | 1365 | 1293 | 1396 | 1075 | $1144.3\pm151.8$ |
| | GAEseq | 1270 | 1121 | 1301 | 1171 | 1245 | 1152 | 1371 | 1105 | 1152 | 1200 | $1208.8\pm85.8$ |
| | CAECseq | 1490 | 1616 | 1347 | 1675 | 1475 | 1405 | 1563 | 1413 | 1436 | 1554 | $1497.4\pm103.1$ |
| | NeurHap | **1029** | **990** | **1097** | **956** | **1012** | **899** | **1014** | **1008** | **1079** | **997** | $\mathbf{1008.1}\pm\mathbf{56.3}$ |
| | Reads | 500 | 500 | 500 | 500 | 500 | 499 | 498 | 497 | 499 | 500 | - |
| | SNPs | 2384 | 2358 | 2385 | 2360 | 2386 | 2383 | 2375 | 2373 | 2353 | 2353 | - |
| 15-strain ZIKV | TenSQR | 941 | 794 | 859 | 869 | 950 | 856 | 848 | 789 | 849 | 758 | $851.3\pm61.5$ |
| | GAEseq | 1470 | 1585 | 1515 | 1590 | 1713 | 1363 | 1523 | 1348 | 1618 | 1393 | $1511.8\pm119.4$ |
| | CAECseq | 2344 | 2248 | 2427 | 2338 | 2454 | 2406 | 2378 | 2496 | 2292 | 2309 | $2369.2\pm77.4$ |
| | NeurHap | **718** | **655** | **752** | **721** | **862** | **658** | **694** | **622** | **666** | **675** | $\mathbf{702.3}\pm\mathbf{67.8}$ |

## 4.3 Visualization

To better understand the phasing results, we use python-iGraph package to visualize the read-overlap graph of Sim-Potato-5X dataset with clustering results from `NeurHap`, CAECseq, and GAEseq (see Figure 3). Different colors denote distinct haplotypes (the number of haplotypes for Sim-Potato is 4). Grey edges are conflicting edges and blue edges are consistent edges in the read-overlap graph. The color of nodes in Figure 3 are derived from the clusters constructed by different models, i.e., each color indicates a cluster of reads that are inferred to come from the same haplotype. In Figure 3 b) and c), 89 and 133 conflicting edges are violated (i.e., connecting two vertices with the same color) for CAECseq and GAEseq, respectively, while none of the conflicting edges are violated for `NeurHap`. `NeurHap` derives a coloring assignment that is most consistent with the conflicting and consistent edges in the read-overlap graph.

Figure 4 shows the search process of `NeurHap` on the Sim-Potato-5X Sample 1 as an example. The sub-figure a) shows the grid layout of the initial coloring of the read-overlap graph violates significant number of conflicting edges (in grey) and consistent edges (in blue). With the increasing number of epochs, the number of violating constraints (conflicting and consistent edges) decrease significantly.

## 4.4 Experimental Analysis

**Ablation study.** To study the effectiveness of our proposed model, we conduct an ablation study to examine the two algorithmic components in `NeurHap`, a graph neural network-based algorithm `NeurHap`-search and a local combinatorial optimisation-based refinement algorithm `NeurHap`-refine.

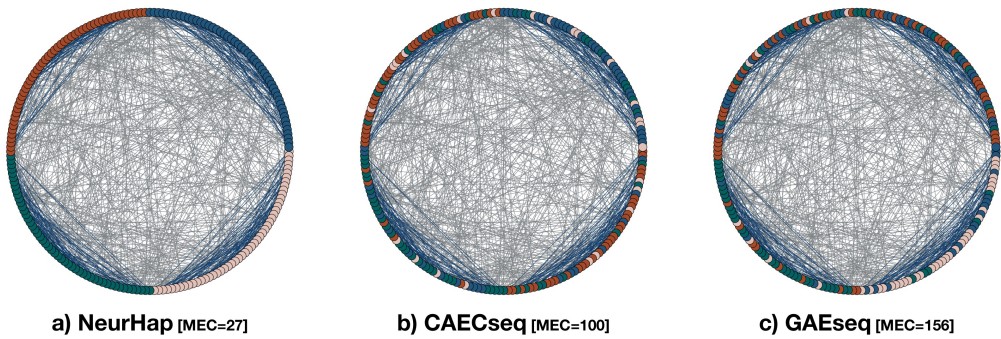

**a) NeurHap** [MEC=27]     **b) CAECseq** [MEC=100]     **c) GAEseq** [MEC=156]

Figure 3: The visualization of `NeurHap`, CAECseq, and GAEseq on Sim-Potato-5X-Sample1 data.

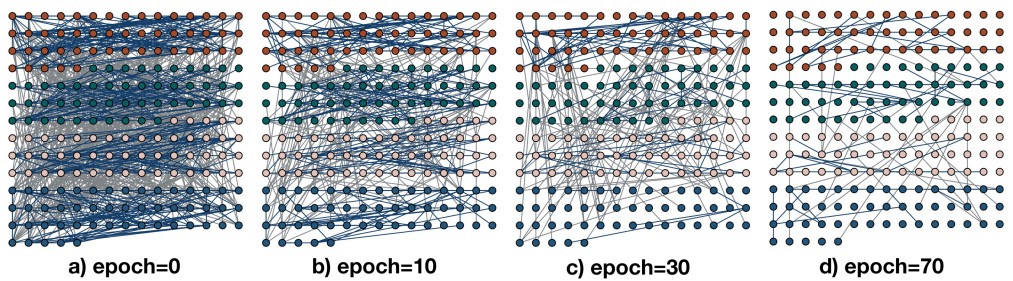

**a) epoch=0**     **b) epoch=10**     **c) epoch=30**     **d) epoch=70**

Figure 4: The grid layout of read-overlap graph with the violating edges in the training of `NeurHap`.

Figure 5 shows that `NeurHap-refine` is able to further optimize the MEC score, e.g., the MEC scores for 5-strain HIV by `NeurHap-search` and `NeurHap` are 1453.5 and 1371.4 respectively, demonstrating the complementary effectiveness of global search and local refinement algorithms on phasing haplotypes.

**Parameter analysis & Running time.** We investigate the importance of core parameters in model, including $p$ and $q$ for read-overlap graph, $\lambda$ for consistent constraints, $t$ for iterations, and $d$ for feature dimension. The detailed parameters analysis is listed in the Appendix A.3. We benchmark the running time of `NeurHap` against two deep learning baselines CAECseq and GAEseq on the Sim-Potato-Cov30 data. `NeurHap` achieves the lowest MEC score (142.0) compared with CAECseq (372.9) and GAEseq (496.9). The running time of `NeurHap` is 258 seconds which is faster than CAECseq (341 seconds). GAEseq is the slowest among the three and takes 492 seconds.

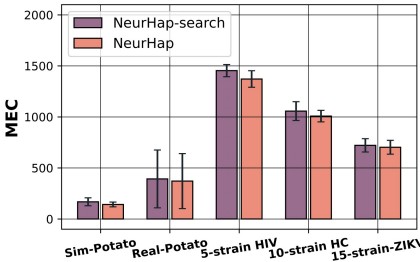

Figure 5: Results of `NeurHap-search` and `NeurHap` (`NeurHap-search` + `NeurHap-refine`) on all five datasets.

## 5   Conclusion

In this paper, we propose `NeurHap`, a graph representation learning approach to reconstruct haplotypes of polyploid species and viral quasispecies. We give a novel formulation of the haplotype phasing problem as a graph coloring problem. We design a message-passing based graph neural network search framework over a carefully constructed graph to assign colors (indicating haplotypes) to the reads, and a local refinement step to adjust colors to optimize MEC scores. Extensive experiments on both simulated and real-world datasets demonstrate the effectiveness of our proposed `NeurHap` model on phasing haplotypes from polyploid species and viral quasispecies. A limitation of our method is in its ability to handle long reads. Massive long reads in polyploids leads to an even larger search space that may be addressed by extensions to our approach in future work. Besides, NeurHap cannot automatically discover the number of haplotypes. This limitation will be addressed going forward.

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
