# Graph Coloring via Neural Networks for Haplotype Assembly and Viral Quasispecies Reconstruction

**Hansheng Xue,**[1] **Vaibhav Rajan,**[2] **and Yu Lin**[1*]

[1]School of Computing, Australian National University, Canberra, Australia
[2]School of Computing, National University of Singapore, Singapore
{hansheng.xue,yu.lin}@anu.edu.au, vaibhav.rajan@nus.edu.sg

## A   Appendix

### A.1   The pseudocode for `NeurHap`-refine algorithm

From the previous `NeurHap`-search step, we obtain an initial coloring assignment for vertices that satisfy the constraints of the read-overlap graph. However, it may exists multiple coloring assignments that satisfy all constraints. Therefore, we run an additional local refinement step to further optimise the MEC score. `NeurHap`-refine mainly searches for possible color adjustments of individual vertices given their associated conflicting and consistent constraints. More specifically, if an individual vertex can be assigned a color different from its current color without violating any of associated conflicting constraints with the neighboring vertices, the color is changed if a better MEC score is obtained by the change. The local refinement algorithm, `NeurHap`-refine, iteratively explores these possible color adjustments of individual vertices. The pseudocode for the `NeurHap`-refine is as follows:

---
**Algorithm 1:** The Local Refinement Algorithm `NeurHap`-refine.

---
**Data:** Read-overlap graph $\mathcal{G}$; number of polyploids $k$; initial color assignment $\mathcal{Y}$
**Result:** final color assignments $\mathcal{Y}^*$.

1 Tag $\leftarrow$ True   // Initialize the iteration tag as True
2 **while** *Tag == True* **do**
3     Tag $\leftarrow$ False   // Set the iteration tag as False
4     **for** *node $v \in \mathcal{V}$* **do**
5        $CN_v \leftarrow \{c(u)|(v,u) \in \mathcal{E}_{\neq}\}$   // Compute the set of colors from conflicting neighbors
6        **for** $c' \notin CN(v)$ *and* $c' \neq c(v)$ **do**
7           // for every possible alternative color $c'$ for $v$
8           $\mathcal{Y}' \leftarrow \mathcal{Y}_{c(v) \leftarrow c'}$
9           // $\mathcal{Y}'$ is derived by setting $c'$ as the color of $v$ in $\mathcal{Y}$
10           **if** $MEC(\mathcal{Y}') < MEC(\mathcal{Y})$ **then**
11              $\mathcal{Y} \leftarrow \mathcal{Y}'$   // Move to a better coloring scheme
12              Tag $\leftarrow$ True   // Set the iteration tag to be true
13           **end**
14        **end**
15     **end**
16 **end**
17 $\mathcal{Y}^* \leftarrow \mathcal{Y}$   // Output the final coloring scheme

---

---
*Corresponding author.

36th Conference on Neural Information Processing Systems (NeurIPS 2022).

## A.2 Implementation Details

Two categories of datasets are used in the paper, *Polyploid species* and *Viral Quasispecies*. *Polyploid species* contains two datasets, Sim-Potato ($k = 4$) and Real-Potato ($k = 4$), which are downloaded from CAECseq [Ke and Vikalo, 2020a] and GAEseq [Ke and Vikalo, 2020b]. *Viral Quasispecies* contains three datasets, 5-strain HIV ($k = 5$), 10-strain HCV ($k = 10$), and 15-strain ZIKV ($k = 15$), which are downloaded from SAVAGE [Baaijens et al., 2017]. It has two steps to generate the SNP matrix, i) Align reads to a reference genome and ii) Extract the matrix from the alignment.

**i) Align Reads to Reference.** BWA-MEM [Li, 2013] is used to align reads to the reference genome. The detailed command is (take the 15-strain ZIKV as an example):

```
$ ./bwa index 15-strain-ZIKV.fasta
$ ./bwa mem 15-strain-ZIKV.fasta forward.fastq reverse.fastq >
15-strain-ZIKV.sam
```

**ii) Extract the SNP Matrix.** We use the same tool described in CAECseq and GAEseq [Ke and Vikalo, 2020a,b] to derive the SNP matrix from the above alignment to ensure a fair comparison. The default parameters are used in the configure file which is same with CAECseq and GAEseq. The detailed command is:

```
$ ./ExtractMatrix config
```

For all five datasts, we randomly generate 10 samples. The detailed number of reads and SNPs for Real-Potato, 5-strain HIV, 10-strain HCV, and 15-strain ZIKV are listed in the paper. For Semi-Potato, sequencing coverage is varied from 5X to 30X. We have 40 sub-datasets in Semi-Potato. The read numbers range from approximately 200 to 1200 and the number of SNPs vary from 200 to 400.

**Read-overlap Graph.** After obtaining the SNP matrix, we build the consistent and conflicting edges between pairs of reads (i.e., pairs of rows in the SNP matrix). Two parameters are introduced in this step to the construct read-overlap graph, $p$ and $q$. Two overlapping reads $R_i$ and $R_j$ are *consistent* if they have the same alleles over all SNP positions meanwhile the length of overlapping is larger than $p$ (i.e., $HD(R_i, R_j) = 0$), and are in *conflict* if they differ on at least $q$ SNP positions (i.e., $HD(R_i, R_j) \geq q$), where $HD(R_i, R_j)$ represents the Hamming distance between two overlapping reads in the read-overlap graph. We adjust two thresholds according different datasets from 2 to 6, and we also evaluate the effect of two parameters for the `NeurHap` model.

You can simply run the following to reproduce the experimental results (we take the Sim-Potato-Cov5 Sample 1 as an example).

```
$ python main.py -e 2000 -t 10 -f 32 -k 4 -r 1e-3 -p 6 -q 2 -l 0.01 -d
Semi-Potato -s Sample1
```

where parameter `-e` represents the number of epoch, `-t` is the number of the iteration, `-f` is the dimension of the embedding, `-k` is the number of haplotypes or ploids, `-r` is the learning rate, `-l` denotes the $\lambda$. Parameters `-d` and `-s` are used to select the corresponding data and sample. The source code of NeurHap is freely available at `https://github.com/xuehansheng/NeurHap`.

**Running environment.** `NeurHap` is implemented in Python 3.6 and Pytorch 1.8 using the Linux server with 6 Intel(R) Core(TM) i7-7800X CPU @ 3.50 GHz, 96GB RAM and 2 NVIDIA RTX A6000 with 48GB memory.

## A.3 Experimental Analysis

**Parameters Analysis.** In this section, we investigate the importance of core parameters in model, including $p$ and $q$ for read-overlap graph, $\lambda$ for consistent constraints, $t$ for iterations, and $d$ for feature dimension. Figure 1 b) shows that our proposed `NeurHap` is robust to the dimension of latent embedding $d$. In Figure 1 a), the MEC scores for `NeurHap` with $\lambda$ varying from 0.0 to 0.1 do not change too much and relatively stable. However, if we choose $\lambda$ as 0.5, the performance of `NeurHap` being worse. We vary $\lambda$ from 0.0 to 0.1 for `NeurHap`.

Next, we investigate the effects of parameters $t$, $p$, and $q$ (take the Sim-Potato-Cov30X Sample 1 as an example). We vary iteration $t$ from 5 to 25 and the results are shown in Figure 2 a). When the iteration $t = 10$, `NeurHap` achieves the best performance on the Sim-Potato-Cov30X Sample 1 dataset. When the iteration $t \geq 10$, the MEC score of the `NeurHap` is relatively stable. In Figure 2 b),

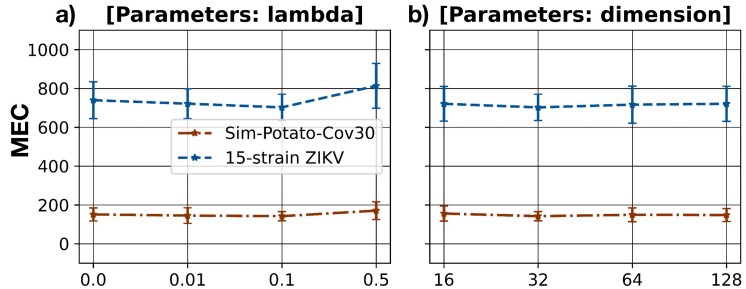

Figure 1: Parameters analysis of the `NeurHap` model ($\lambda$ and $d$).

When parameters $q = 4$ and $p = 5$, the `NeurHap` achieves the best performance. If the parameter $p < 5$ ($q$ is fixed to 4), the MEC score of the `NeurHap` is high because the constructed consistent edges are not confident and they contain several mistaken consistent edges. When the parameter $q > 4$ (the $p$ is fixed to 5), the number of extracted conflicting edges is few (the read-overlap graph is sparse) which is not good to optimise the MEC score.

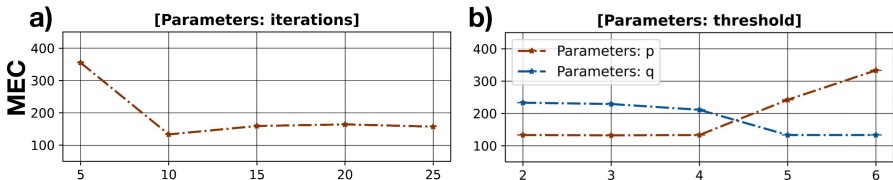

Figure 2: Parameters analysis of the `NeurHap` model ($t$, $p$, and $q$).

**Running Time.** We benchmark the running time of `NeurHap` against two deep learning baselines CAECseq and GAEseq on the Sim-Potato-Cov30 Sample 1 data. NeurHap achieves the lowest MEC score (142.0) compared with CAECseq (372.9) and GAEseq (496.9). The running time of NeurHap is 258 seconds which is faster than CAECseq (341 seconds). GAEseq is the slowest among the three and takes 492 seconds.

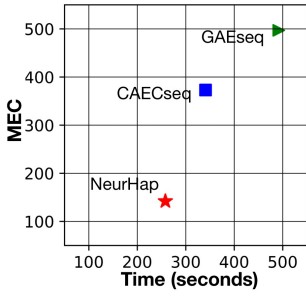

Figure 3: The running time of `NeurHap`, CAECseq and GAEseq.

## A.4 Additional Experiment

**Average MEC on Semi-Potato.** In the experimental part, we select the lowest MEC score as the final results after running experiment 10 times which is same as previous SOTA baselines Ke and Vikalo [2020a,b]. Here, we also report the average MEC score after running all algorithms on Semi-Potato 10 times (see Table 1). `NeurHap` still outperforms other SOTA baselines.

**Benchmark against Graph Coloring.** We also benchmark `NeurHap` against two graph coloring algorithms, including Greedy [Brélaz, 1979] and RUN-CSP [Toenshoff et al., 2019]. We implement graph coloring algorithms on the read-overlap graphs which only contain conflicting edges because those methods cannot address the consistent edges. In Table 2, `NeurHap` significantly outperforms graph coloring algorithms.

Table 1: Performance comparison on Sim-Potato (Tetraploid, $k = 4$).

| Polyploids | Model | #Cov 5X | #Cov 10X | #Cov 20X | #Cov 30X |
|---|---|---|---|---|---|
| Tetraploid ($k=4$) | H-PoP | 429.0±64.1 | 933.9±103.6 | 1782.2±161.8 | 2826.9±180.7 |
| | AltHap | 610.9±259.3 | 722.3±179.1 | 649.3±369.4 | 1148.2±509.9 |
| | GAEseq | 225.1±17.7 | 391.2±45.5 | 610.4±97.3 | 811.8±131.8 |
| | CAECseq | 160.5±25.9 | 266.0±43.3 | 466.5±89.0 | 629.5±160.0 |
| | NeurHap | **37.5±5.5** | **62.8±7.5** | **113.2±19.8** | **166.3±26.7** |

Table 2: Performance comparison on Real-Potato ($k = 4$) and 5-strain HIV ($k = 5$).

| Data | Model | # 1 | # 2 | # 3 | # 4 | # 5 | # 6 | # 7 | # 8 | # 9 | # 10 | # Avg. |
|---|---|---|---|---|---|---|---|---|---|---|---|---|
| Real-Potato | Greedy | 296 | 458 | 162 | 3 | 239 | 1014 | 679 | 602 | 694 | 906 | 505.3±330.3 |
| | RUN-CSP | 186 | 358 | 107 | 1 | 185 | 890 | 553 | 492 | 647 | 767 | 418.6±298.7 |
| | NeurHap | **178** | **343** | **93** | **1** | **163** | **857** | **499** | **384** | **561** | **632** | **371.6±268.9** |
| 5-strain HIV | Greedy | 3974 | 3791 | 3633 | 3819 | 4251 | 3472 | 3137 | 3241 | 3326 | 3476 | 3612.0±349.3 |
| | RUN-CSP | 2226 | 2375 | 2175 | 2408 | 2192 | 2748 | 2449 | 2614 | 2312 | 2567 | 2406.6±191.3 |
| | NeurHap | **1307** | **1525** | **1385** | **1265** | **1410** | **1382** | **1393** | **1323** | **1274** | **1450** | **1371.4±81.2** |

**MEC score v.s. Violating Constraints.** While Eqn. 2 aims to minimize the sum of hamming distances between each read $\mathcal{R}_j$ and the haplotype $\mathcal{H}_i$ that is drawn from $\mathcal{R}_j$, Eqn. 3 aims to minimize the divergence between pairs of reads (as $P(v_i)$ and $P(v_j)$) that are drawn from the same haplotype and maximize the divergence between pairs of reads if they are drawn from different haplotypes. Moreover, the hamming distances in Eqn.2 have been used implicitly to derive in Eqn.3. In an ideal case, if all pairs of conflicting reads are assigned into different haplotypes (i.e., different colors) and all pairs of consistent reads are assigned into the same haplotypes (i.e., the same color), each cluster will only contain consistent reads and thus the MEC score in Eqn.2 will be minimized to be 0. In non-ideal cases such as Sim-Potato Cov-5X Sample 1 datasets, the following Figure 4 shows that the objective function to be minimized in Eqn.2 (i.e., MEC) correlates well with the objective function to be minimized in Eqn.3, which demonstrates the effectiveness of NeurHap for minimizing the MEC through optimizing Eqn.3.

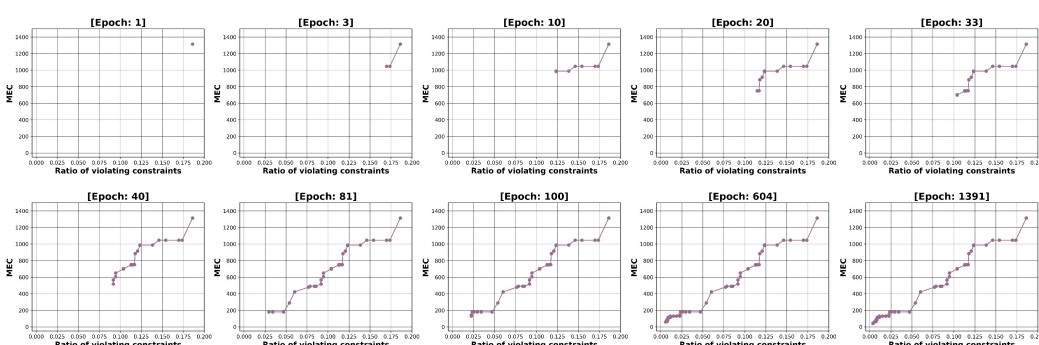

Figure 4: The MEC score v.s. Violating Constraints in the training process of `NeurHap`.

**Scalability.** To evaluate the scalability of the NeurHap, we incrementally combine the samples in Real-Potato dataset and summarise the results in Table 3. It is observed that the running time of NeurHap is roughly linearly correlated with the number of total edges (conflict edges + consistent edges). On the other hand, diploid haplotype assembly remains challenging for reconstructing chromosome-level haplotypes, especially for large eukaryotic genomes with complex repeats. Similar to CAECseq and GAEseq, NeurHap has also focused on short-read datasets on gene regions because complex repeats in the intergenic regions along the chromosome make it impossible to reconstruct continuous haplotypes reliably.

Besides, we applied NeurHap on a chromosome-level dataset for Chromosome 22 of the human genome to validate the scalability of NeurHap. Specifically, we downloaded publicly

Table 3: Performance comparison on cumulative Real-Potato dataset.

| Samples | | # 1 | # 2 | # 3 | # 4 | # 5 | # 6 | # 7 | # 8 | # 9 | # 10 |
|---|---|---|---|---|---|---|---|---|---|---|---|
| Reads | | 240 | 629 | 903 | 1018 | 1159 | 1557 | 1852 | 2136 | 2625 | 3074 |
| SNPs | | 295 | 533 | 616 | 639 | 815 | 1013 | 1469 | 1893 | 2129 | 2539 |
| Conflict | | 3351 | 6380 | 13433 | 14208 | 16207 | 34288 | 39368 | 42811 | 51358 | 59621 |
| Consistent | | 966 | 1514 | 3323 | 3537 | 3908 | 5747 | 6470 | 6977 | 7985 | 9329 |
| CAECseq | MEC | 229 | 786 | 910 | 985 | 1282 | 1997 | 2584 | 3018 | 3914 | 4524 |
| | time | 243s | 283s | 302s | 310s | 414s | 586s | 798s | 1188s | 1991s | 2774s |
| NeurHap -search | MEC | 183 | 559 | 671 | 692 | 888 | 1802 | 2305 | 2667 | 3316 | 3992 |
| | time | 28s | 38s | 52s | 53s | 63s | 99s | 128s | 157s | 200s | 253s |

available alignment files for the Human Genome NA12878 (from `http://s3.amazonaws.com/nanopore-human-wgs/NA12878-Albacore2.1.sorted.bam`) and combined them with the set of heterozygous SNPs on Chromosome 22 of the human genome (derived from [Duitama et al., 2012]) to build the input alignment matrix (following the same procedure introduced in HapCUT2 [Edge et al., 2017]). This constructed matrix contains 129,338 long reads and 22,792 SNPs. NeurHap took 734 seconds and around 13G memory to reconstruct two chromosome-level haplotypes with a MEC score of 23,114. As Chromosome 22 is about 1.6% of the whole human genome and 20% of the largest chromosome (Chromosome 1) in the human genome, we estimate (optimistically) that phasing all the chromosomes in the human genome will take about 12 hours with a peak memory of 65G. Note that CAECseq and GAEseq are both out of the memory when they were applied to this chromosome-level dataset on the NVIDIA RTX A6000 with 48GB memory.