# OpenReview forum: "Graph Coloring via Neural Networks for Haplotype Assembly and Viral Quasispecies Reconstruction"
_NeurIPS.cc/2022/Conference — NeurIPS 2022 Accept_

### Official Review · Reviewer_BGfU · 2022-07-10

**Rating:** 6
**Confidence:** 3
**Soundness:** 3 good
**Presentation:** 2 fair
**Contribution:** 3 good

**Summary:**

Haplotype assembly is crucial for resolving the fine structure of structural variation, detecting copy number variation, SNP panel imputation (these tasks in diploids), and reconstruction of viral "quasispecies". Assembly is more difficult with shallower sequencing, increasing ploidy, increasing sequencing error, and increasing numbers of related strains.

The authors define the assembly problem for ploidy > 2, given a priori a number N of chromosomes or species, and a set of reads aligned to a reference, as assigning reads to N haplotypes ("colouring") are aligned to minimize the summed hamming distances between each of the reads and their respective consensus haplotype (the so-called MEC). This problem is unfortunately NP-hard.

The authors first relax the problem by allowing themselves to merely assign each read a probability of belonging to each haplotype, solving the original problem by simply assigning each read the most likely colour (this is taken from previous work). Next they write a heuristic loss for this relaxed problem. The loss penalizes overlapping and consistent (> p bases overlapping, all the same) reads being assigned to different haplotypes and overlapping and inconsistent (> p bases overlapping, > q different - p, q are hyperparameters) being assigned to the same haplotype.

Precomputing which reads overlap and are consistent or inconsistent, the authors propose minimizing the objective the following way:
assign each read an embedding that is put through a neural network
After obtaining a colouring by minimizing this heuristic objective, the authors "tune" the colouring to minimize the MEC as well. However, in figure 5 (compared to table 3), the authors demonstrate that most of the performance improvement of the model compared to previous methods comes from simply minimizing the heuristic loss.

The authors demonstrate their ability to minimize the MEC on a number of benchmarks

**Questions:**

1) My primary question: The loss in equation (3) must be minimized over a space of dimension #reads X (#haplotypes-1). The largest this dimension ends up being in the benchmarks is 500 x 14 = 7000. Is it not thus possible to perform gradient descent on the haplotype membership probabilities directly? It also seems that since the GNN method involves optimizing the initial embeddings of all reads, optimizing the GNN is a higher dimensional problem than optimizing the membership probabilities directly. (Other NN methods cited -CAECseq and GAEseq- avoid the problem of the dimensionality of the optimization problem scaling with number of reads by using the one-hot encoded sequence or the column of the SNP matrix to construct the initial embedding, in principle reducing the dimension of the optimization problem.)

2) Why do the benchmarks contain so few reads and only aim to reconstruct small parts of a chromosome? Methods exist for diploid haplotype assembly that are accurate for 100s of kb containing substantially more than 1000 reads. H-PoP looked to assemble over 60 thousand reads with 12 thousand SNPs in their publication. Is scaling to the size of an entire chromosome a difficulty for your method? Is this scaling not necessary for use of these algorithms?

3) Why only focus on MEC as a performance metric? It is known that the presence of reads with errors means that the correct phasing may not be the one with the lowest MEC. Many alternatives, including switch error rate are presented in the author's citation Saada OA et al, 2022. This seems to be an important consideration especially when introducing a new metric to minimize.

**Limitations:**

It seems that the primary limitation of this method is its inability to scale beyond small datasets assembling short reads from only a fraction of a chromosome at once. At least evidence was not shown of scaling to larger datasets.
The authors have not suggested a method to tune hyperparameters p and q for arbitrary datasets, which may be required for sequencing with different read length and quality.

**Strengths And Weaknesses:**

Strengths:
1) Data and code was easily accessible, making it easy to tell what the authors did. Although the particular parameters used to run the code are not available.
2) The authors seemingly pose what seems to be a new objective for the relaxed assembly problem whose minimization seems to also minimize the MEC.
3) The authors propose a method that, in attempting to minimize this new objective, succeeds in minimizing the MEC more than previous methods.
Weaknesses:
1) I am concerned the benchmarks do not justify the use of the GNN.
2) I am concerned benchmarking using only MEC on very small datasets may not display any possible pathology of the assembly and doesn't demonstrate the ability of the model to scale to relevant data sizes.
(See questions)
Because of these concerns, the validation provides enough guidance to practitioners on the strengths and efficacy of the model.
3) The main text leaves certain concepts ambiguous: i) the authors did not make it clear in the main text what parameters were being optimized in the methods, in particular the initial embeddings. ii) it was not stated unequivocally that the authors themselves had come up with the objective (3), or what they were inspired by.

---

> ### Author Response · Authors · 2022-08-02
> **Author Responses to Reviewer BGfU**
>
> Thanks for the comments.
>
> Q1: Yes, it is possible to perform gradient descent to optimize the loss in Eq. 3 directly. The results are not as good as GNN-based approaches as shown in the table below.
>
> In NeurHap, we use the trainable embedding as the initial input and the dimension size in NeurHap was set as 32. Here we perform more experiments on the Real-Potato dataset to explore other encoding scenarios, including the one-hot encoding (the same as CAECseq), the SNP matrix encoding (the same as GAEseq), and the Color-Dim encoding (as suggested above, using \#haplotypes). We find that the NeurHap (trainable initial embedding) achieves the overall best performance among all encoding scenarios. Trainable initial node embedding can accelerate the training and is not much different from other initialization methods, such as one-hot encoding, adjacency matrix, degree et al. If we use the number of haplotypes to set up the embedding size, the scale of training gets smaller which makes GNN-based models more difficult to be trained. Performing gradient descent to optimize the loss function and initial embedding is possible. However, it cannot capture the typology of the graph and optimize the haplotypes probability in a low-dimensional space. In contrast, GNN can effectively learn the structural information of the graph in a non-linear high-dimensional space. We will add these results in the revised version of our paper.
>
> |Data| #1 | #2 | #3|  #4 | #5 | #6 | #7 | #8 | #9 | #10 | Avg.|
> | :--------  | :-----  | :----:  | :--------  | :-----  | :----:  |:----:  |:-----  | :----:  |:----:  |:-----  | :----:  |
> | NeurHap | 178 | 343 | 93 | 1 | 163 | 857 | 499 | 384 | 561 | 632 | 371.6 |
> |NeurHap+Onehot | 183 | 368 | 106 | 1 | 189 | 868 | 489 | 415 | 615 | 672 | 390.6 |
> | NeurHap+SNPmatrix | 191 | 375 | 106 | 1 | 169 | 876 | 497 | 389 | 597 | 655 | 385.6 |
> | NeurHap+ColorDim | 206 | 439 | 93 | 1 | 188 | 1007 | 532 | 385 | 660 | 713 | 422.4 |
> | Gradient Descent | 497 | 572 | 404 | 70 | 329 | 1368 | 732 | 714 | 802 | 1058 | 654.6 |
>
> Q2: To evaluate the scalability of the NeurHap, we incrementally combine the samples in Real-Potato dataset and summarise the results in the following table. It is observed that the running time of NeurHap is roughly linearly correlated with the number of total edges (conflict edges + consistent edges). On the other hand, diploid haplotype assembly remains challenging for reconstructing chromosome-level haplotypes, especially for large eukaryotic genomes with complex repeats. Similar to CAECseq and GAEseq, NeurHap has also focused on short-read datasets on gene regions because complex repeats in the intergenic regions along the chromosome make it impossible to reconstruct continuous haplotypes reliably. As NeurHap has demonstrated its potential scalability, in future we will investigate whether NeurHap can be applied to long-read datasets to reconstruct chromosome-level haplotypes.
>
> |Samples| | #1 | #2 | #3|  #4 | #5 | #6 | #7 | #8 | #9 | #10 |
> | :--------  | :-----  | :----:  | :--------  | :-----  | :----:  |:----:  |:-----  | :----:  |:----:  |:-----  | :----:  |
> | Reads | | 240 | 629 | 903 | 1018 | 1159 | 1557 | 1852 | 2136 | 2625 | 3074 |
> | SNPs | | 295 | 533 | 616 | 639 | 815 | 1013 | 1469 | 1893 | 2129 | 2539 |
> | Conflict | | 3351 | 6380 | 13433 | 14208 | 16207 | 34288 | 39368 | 42811 | 51358 | 59621 |
> | Consistent | | 966 | 1514 | 3323 | 3537 | 3908 | 5747 | 6470 | 6977 | 7985 | 9329 |
> | CAECseq | MEC | 229 | 786 | 910 | 985 | 1282 | 1997 | 2584 | 3018 | 3914 | 4524 |
> | | Time | 243s | 283s | 302s | 310s | 414s | 586s | 798s | 1188s | 1991s | 2774s |
> | NeurHap | MEC | 183 | 559 | 671 | 692 | 888 | 1802 | 2305 | 2667 | 3316 | 3992 |
> | -search | Time | 28s | 38s | 52s | 53s | 63s | 99s | 128s | 157s | 200s | 253s |
>
> Q3: We follow the pipeline and evaluation metric of previous state-of-the-art models, CAECseq and GAEseq. Haplotype phasing can be grouped into different categories according to the optimization object, such as minimizing SNP removal, maximizing fragments cut, minimizing error correction (MEC) score etc. Most haplotype reconstruction approaches developed in recent years focus on optimizing the MEC score. Our method also belongs to this category (optimizing MEC score). Because of the time limitation, we do not explore other evaluation metrics this time and will introduce more metrics in the future.

---

> > ### Comment · Reviewer_BGfU · 2022-08-03
> > **Response to the authors**
> >
> > I thank the authors for their thorough response.
> >
> > Q1: It is interesting that gradient descent performs poorly. Could the authors briefly clarify why they think this is the case? Is it falling into local minima? If so, then the use of a GNN can be justified as a way to make the loss landscape easier to optimize.
> >
> > Q2: Could you clarify if the relationship between these dataset sizes and the scalability required for full chromosome reconstruction? In particular, is it possible that a method like this could, in principle, scale to the whole genome, especially since the number of parameters scales with the number of reads? Would it be correct to say that there exist substantially more scalable methods such as HPoP?
> >
> > Q3: It remains a weakness of the paper that assembly is benchmarked using only a single metric.
> >
> > My weakness labelled (6) remains unaddressed.

---

> > > ### Author Response · Authors · 2022-08-06
> > > **Author Responses to Reviewer BGfU**
> > >
> > > Thanks for the comments.
> > >
> > > Q1: The objective function in Eq.3 aims to optimise the haplotypes' probability and minimise the violation of two constraints, conflict $E_{\neq}$ and consistent $E_{=}$. The matrix $M$ in Eq.3 denotes the binary relationships among $k$ colors ($M_{=}$ for consistent and $M_{\neq}$ for conflict relationships). Directly optimising haplotypes probability via gradient descent treats reads as independent nodes and is likely to fall into local minima because it cannot make full use of the global structure of the read-overlap graph. On the other hand, the GNN solver captures all the relationships among reads in the read-overlap graph simultaneously and encodes reads into a high dimensional space (where the number of dimensions is greater than the number of haplotypes). The GNN learning model can model the implicit relations among reads which is essential in the graph colouring optimization problem.
> > >
> > > Q2: In our previous rebuttal, we incrementally combine 10 samples in the Real-Potato dataset to increase the number of reads and SNPs. The running time of NeurHap is roughly linearly correlated with the number of edges in the read-overlap graph. Therefore, we believe that NeurHap can scale up to work at the whole genome level. As we were not able to find the chromosome-level dataset used by HPoP, we applied NeurHap on a dataset for Chromosome 22 of the human genome to validate the scalability of NeurHap. Specifically, we downloaded publicly available alignment files for the Human Genome NA12878 (from http://s3.amazonaws.com/nanopore-human-wgs/NA12878-Albacore2.1.sorted.bam) and combined them with the set of heterozygous SNPs on Chromosome 22 of the human genome (derived from [1]) to build the input alignment matrix (following the same procedure introduced in HapCUT2 [2]).
> > > This constructed matrix contains 129,338 long reads and 22,792 SNPs. NeurHap took 734 seconds and around 13G memory to reconstruct two chromosome-level haplotypes with a MEC score of 23,114. As Chromosome 22 is about 1.6\% of the whole human genome and 20\% of the largest chromosome (Chromosome 1) in the human genome, we estimate (optimistically) that phasing all the chromosomes in the human genome will take about 12 hours with a peak memory of 65G. Note that CAECseq and GAEseq are both out of the memory when they were applied to this chromosome-level dataset on the NVIDIA RTX A6000 with 48GB memory.
> > >
> > > Q3: Following CAECseq, we further use a new metric (correct phasing rate, CPR) to evaluate the phasing accuracy when the ground-truth is known, in addition to the classic MEC metric without knowing the ground-truth. According to CAECseq, this new metric CPR is computed as
> > > \begin{equation*}
> > >     CPR = 1- \frac{1}{k\times n}(\min\sum_{i=1}^{k} HD(H_i,F(H_i)))
> > > \end{equation*}
> > > where $k$ and $n$ denote the number of haplotypes and SNPs, $H$ is the reconstructed haplotypes, $HD$ is the hamming distance, $F$ is the best one-to-one mapping from the reconstructed haplotypes to ground-truth haplotypes.
> > >
> > > We took the real HIV-1 data as an example to to validate the CPR performance of NeurHap and CAECseq. The following table shows both the MEC and CPR scores of NeurHap and CAECseq on four genes in the real HIV-1 data (NeurHap and CAECseq both achieve the same highest CPR score (1.0) on the other nine genes). Note that NeurHap outperforms CAECseq on the real HIV-1 data in terms of the CPR score.
> > >
> > > |Models | | gp120 | vpu | gp41 | nef |
> > > | :--------  | :-----  | :----:  | :--------  | :-----  | :----:  |
> > > | NeurHap | MEC | 55894 | 84619 | 9571 | 35953 |
> > > | | CPR | 0.9982 | 1.0 | 0.9976 | 0.9926 |
> > > | CAECseq | MEC | 70486 | 89413 | 9387 | 35956 |
> > > | | CPR | 0.9836 | 0.9954 | 0.9968 | 0.9923 |
> > >
> > > | | gp120 | vpu | gp41 | nef |
> > > | :--------  | :-----  | :----:  | :--------  | :-----  |
> > > | Number of reads | 69534 | 33747 | 62428 | 23697 |
> > > | Number of SNPs | 215 | 35 | 132 | 89 |
> > >
> > >
> > > [1] J. Duitama, et al., ‘’Fosmid-based whole genome haplotyping of a hapmap trio child: evaluation of single individual haplotyping techniques,” Nucleic acids research, vol. 40, no. 5, pp. 2041–2053, 2011.
> > >
> > > [2] P.  Edge, et al., ‘’Hapcut2:  robust and accurate haplotype assembly for diverse sequencing technologies," Genome Research, vol. 27, no. 5, pp. 801–812, 2017

---

> > > > ### Comment · Reviewer_BGfU · 2022-08-07
> > > > **Response to authors**
> > > >
> > > > I thank the authors for their thorough response. This response motivates their method well and demonstrates the scalability of their model. They also address my concern that the metric they use to validate their models may not detect some failure modes by validating with another metric. The authors other response clears up some other minor confusions I had. I increase my soundness and contribution scores to 3 and my overall rating to 6.

---

> > > ### Author Response · Authors · 2022-08-06
> > > **Author Responses to Reviewer BGfU**
> > >
> > > Q4: i) The main parameters to be optimized in the NeurHap model contains the initial embeddings for reads, the MLP's weights in the learning model, and the parameters in the decoder part (haplotypes' probability).
> > > ii) The objective function of Eq. 3 belongs to the NeurHap-search inspired by RUN-CSP (refer to line 195). Eq.3 contains two parts. The first part is to measure the haplotypes' probability with respect to conflicting constraints which is the same as the previous RUN-CSP. The second part is to model the consistent conditions, which differ from RUN-CSP. RUN-CSP only consider the conflicting constraints meanwhile NeurHap consider both the conflicting and consistent constraints.
> > > Besides, our model is simpler than RUN-CSP.
> > > We use simple and effective MLP to capture the topology information instead of a RNN (LSTM or GRU)-based model used in RUN-CSP as the learning module.
> > >
> > > Refer to the following table for a detailed comparison between RUN-CSP and NeurHap-search on Real-Potato and 5-strain HIV. We previously show the MEC scores of NeurHap and RUN-CSP in the supplementary material A.4, and we also add the running time here.
> > >
> > > |Data|Model | | #1 | #2 | #3|  #4 | #5 | #6 | #7 | #8 | #9 | #10 |
> > > | :--------  | :-----  | :----:  | :--------  | :-----  | :----:  |:----:  |:-----  | :----:  |:----:  |:-----  | :----:  |:----:  |
> > > | Real-Potato | RUN-CSP | MEC |186 |358 |107 | 1 | 185 | 890 | 553 | 492 | 647 | 767 |
> > > | | | Time | 36s | 59s | 63s | 53s | 49s | 61s | 47s | 49s | 67s | 58s |
> > > | | NeurHap | MEC | 183 | 363 | 93 | 1 | 172 | 889 | 532 | 399 | 595 | 683 |
> > > | | -search| Time | 28s | 33s | 37s | 39s | 36s | 39s | 36s | 34s | 39s | 40s |
> > > | 5-strain HIV | RUN-CSP | MEC | 2226 | 2375 | 2175 | 2408 | 2192 | 2748 | 2449 | 2614 | 2312 | 2567 |
> > > | | | Time | 408s | 393s | 361s | 365s | 382s | 416s | 368s | 370s | 366s | 369s |
> > > | | NeurHap | MEC | 1436 | 1525 | 1536 | 1346 | 1479 | 1448 | 1460 | 1430 | 1380 | 1495 |
> > > | | -search| Time | 72s | 69s | 71s | 70s | 73s | 74s | 71s | 72s | 69s | 74s |

---

> ### Author Response · Authors · 2022-08-02
> **Author Responses to Reviewer BGfU**
>
> Q4: As discussed in the reponse answer to Q2, we follow the experimental setup in CAECseq and GAEseq, and focus on gene regions from large chromosomes. As NeurHap has demonstrated its potential scalibility, how to apply NeurHap on long-read datasets beyond a fraction of a chromosome will be an interesting direction to explore in the future. Our project can be seen as an exploration to combine neural networks, combinatorial optimization, and haplotype reconstruction. Extensive experiments demonstrate substantially improved performance of NeurHap in real and synthetic datasets compared to competing approaches. NeurHap is faster than other state-of-the-art neural networks-based approaches. Besides, we also explore the possibility of implementing NeurHap on the large dataset.
>
> Two parameters are used in the NeurHap to constrauct the read-overlap graph, $p$ and $q$. One method to tune these parameters is to use grid search iteratively and repeatedly run NeurHap with different combination of parameters. Here we also show that, although the MEC score achieved by NeurHap changes if the parameter $p$ and $q$ are adjusted, NeurHap still outperforms CAECseq, the state-of-the-art baseline. We take Real-Potato Sample 1 as the example and adjust the value of parameter $p$ and $q$ in the range of 1 and 10 (see the following table).
>
> |Param. $p$ | | 1 | 2 | 3|  4 | 5 | 6 | 7 | 8 | 9 | 10 |
> | :--------  | :-----  | :----:  | :--------  | :-----  | :----:  |:----:  |:-----  | :----:  |:----:  |:-----  | :----:  |
> | NeurHap | Num. of Consistent | 1668 | 1349 | 1168 | 1096 | 1019 | 966 | 924 | 885 | 810 | 751 |
> | |MEC | 206 | 182 | 194 | 183 | 169 | 173 | 180 | 187 | 199 | 187 |
> | Param. $q$ | | 1 | 2 | 3 | 4 | 5 | 6 | 7 | 8 | 9 | 10|
> | NeurHap | Num. of Conflict | 4971 | 4123 | 3351 | 2727 | 2212 | 1816 | 1441 | 1030 | 698 | 455 |
> | | MEC| 217 | 188 | 178 | 175 | 191 | 193 | 195 | 190 | 225 | 213 |
> | CAECseq | | 229| 229| 229| 229| 229| 229| 229| 229| 229| 229|

---

### Official Review · Reviewer_X2sa · 2022-07-11

**Rating:** 7
**Confidence:** 4
**Soundness:** 3 good
**Presentation:** 3 good
**Contribution:** 3 good

**Summary:**

This paper explores the haplotype assembly/phasing problem through the lens of graph coloring and proposes a solution based on graph representation learning with a local search refinement. The problem admits a unique graph coloring formulation that differs from conventional graph coloring in that there can be consistent and conflict edges, whereas, traditionally, graph coloring would only handle conflict edges. Similarly, graph neural networks typically assume that similar nodes should have similar embeddings, but this assumption is challenged in the current setting. The authors formulate a loss function to account for both types of edges and their experiments demonstrate a significant improvement over competing approaches.

**Questions:**

1. Can you provide more details on how the COMBINE, AGGREGATE, and MESSAGE functions are defined?
2. How do you see the notions of similar and dissimilar embeddings via consistent and conflict relationships being generalized to graph neural networks more broadly?

**Limitations:**

The authors claim that Section 5 covers limitations, though details are scarce.

Some minor comments:
"haploytypes" -> haplotypes
"Neural Networks on Graph." -> Neural Networks on Graphs.
"functionm" -> function
"} respectively" -> }, respectively
"Two overlapping read are called" -> reads
"naturally motives" -> naturally motivates
"constraints satisfying problem" -> constraint satisfaction problem



**Strengths And Weaknesses:**

Strengths
* A nice reduction from the haplotype assembly/phasing problem to a modified graph coloring problem is proposed, thereby introducing opportunities to leverage work done in that field.
* A novel loss function is derived that accounts for similarity and dissimilarity of vertex relationships via consistent and conflict edges, respectively. This may even be generalized to graph neural networks more broadly.
* Experiments are performed on synthetic a real-world datasets, demonstrating an improvement over competing approaches.

Weaknesses
* Some details are missing on how the COMBINE, AGGREGRATE, and MESSAGE functions are defined

---

> ### Author Response · Authors · 2022-08-01
> **Author Responses to Reviewer X2sa**
>
> Thanks for the comments.
>
> Q1: Different from other graph neural network models with homophily assumption, the information of linked edges among the read-overlap graph is heterophilous (directed linked nodes labelled distinct colors). Thus, in the topology-learning module of the NeurHap-search, we propose a simple message-passing based learning model which mainly focus on the local structural informaion.
> $\mathrm{COMBINE}(\cdot)$ is a combine function and $\mathrm{AGGREGATE(\cdot)}$ denotes an aggregation function. The combine and aggregate operators have multiple choices in the classical graph learning models, such as mean, RNN, attention etc. [1, 2]. To search for a simple model, we adopt most recent message updater and mean operator ($\mathrm{m}(v_i)=\frac{1}{|N(v_i)|}\sum_{v_j\in N(v_i)}\mathrm{h}(v_j)$) as combine and aggregate function respectively.
> $\mathrm{MESSAGE(\cdot)}$ represents the learnable message function, which aims to learn the local topological information of the read-overlap graph e.g. $\mathrm{MESSAGE}(\mathrm{h}(v_i),\mathrm{h}(v_j))=\mathrm{MLP}(\mathrm{h}(v_i)||\mathrm{h}(v_j))$. Two linear layers with activation function are selected to construct the multilayer perceptron (MLP) layer. we will add more details in the paper.
>
> [1] GraphSAGE: Inductive Representation Learning on Large Graphs, W.L. Hamilton et al., NeurIPS'17.
>
> [2] TGNs: Temporal Graph Networks for Deep Learning on Dynamic Graphs, Emanuele Rossi et al., ICML 2020 Workshop.
>
>
> Q2: The topology of the underlying graph in our case is based on consistent and conflict relations. Thus, the learnt embeddings should be faithful to these relations, i.e., if there is a conflicting edge between two nodes, their representations will be more dissimilar (in the representation space) compared to those with no conflicting edges. On the other hand, if there is a consistent edge between two nodes, their representations will be more similar (in the representation space) compared to those with no consistent edges.
>
>
> Q3: Presently we have mentioned the limitation with respect to handling long reads. As suggested by Reviewer 1, we will add more details about the method’s inability to take into account statistical uncertainty, and inability to automatically discover the number of haplotypes (as may be necessary in studying viral quasispecies).
>
>
> Q4: We will fix these typos.

---

> > ### Comment · Reviewer_X2sa · 2022-08-06
> > **Thank you for the thoughtful response.**
> >
> > Thank you for the thoughtful response.

---

### Official Review · Reviewer_uUcF · 2022-07-11

**Rating:** 4
**Confidence:** 4
**Soundness:** 3 good
**Presentation:** 2 fair
**Contribution:** 2 fair

**Summary:**

The paper proposes a GNN-based approach for clustering reads (short biological sequences) that come from a stochastic process of sampling, with errors, subsequences from a few similar, but different, long sequences (e.g. genomes of several viral quasispecies); the ultimate goal of the method is to infer these long sequences from a set of short reads.

**Questions:**

Please comment on the assumptions/design choices (see above), and on the novelty of the method.

**Limitations:**

Limitations are described adequately.

**Strengths And Weaknesses:**

The presentation of the problem and the method is mostly clear, although some design choices are presented without much motivation, and seem somewhat problematic. Specifically:
- why use only SNP positions in determining that reads are overlapping (line 156 and 174)? Why not use positions in which nucleotides are the same (non-SNP positions?). E.g. if two 200bp reads overlap on 100 bases, but are from a region (e.g. a highly conserved, essential gene) that contains very few SNPs (below $p$ SNPs), why are they not considered overlapping?
- In lines 155, one reads “any two reads (…) must be (…) non-overlapping, consistent, conflict. But Consistent is defined as 0 differences in SNPs, Conflict as $\geq q$ differences in SNPs, then two overlapping reads with e.g. $q-1$ differing SNPs are neither Consistent nor Conflicting.
- the requirement that “any two conflicting reads have two different colors and any two consistent reads have the same color” (line 185) seems too strong. What about e.g. three quasispecies GGGGAA, AACCTT, AACCGG and a set of 3 reads AACC CCTT CCGG, p=2, q=2? All three reads are overlapping, and 1 is consistent with 2 & 3, but 2 & 3 are in conflict.

The paper involves a GNN message passing approach for solving the problem of assigning reads to quasispecies (colors), modeled as minimizing the loss (eq. 3) composed of two terms involving inner products of color assignments $p_i$ for vertices $i$: the “consistent” term that is the sum of $-\log( <p_i, p_j> )$ over all pairs $(i,j)$ linked by consistent edge, and the “conflict” term that is the sum of $-log ( 1 - <p_i, p_j> )$ over all pairs $(i,j)$ linked by conflict edge. This is a straightforward extension of cited prior work that only used one edge type and had a single term. Thus, the novelty of the approach on the GNN side is limited.

Finally, the toy example in Figure 2 seems too simple: judging from it, the problem can be solved by finding connected components in the graph with only blue (Consistent) edges taken into account.

---

> ### Author Response · Authors · 2022-08-01
> **Author Responses to Reviewer uUcF**
>
> Thanks for the comments.
>
> Q1: Thanks for pointing out that the term 'non-overlapping' may not convey the relationship we want to express. For better clarity, we think the term 'non-overlapping' could be revised to 'ambiguous' to mean that there is not enough evidence to support that these two reads should belong to the same haplotype ('consistent') or should belong to the different haplotypes ('conflict'), even when two reads are overlapping (i.e., spanning over common genomic regions with or without SNPs).
>
> Therefore, in the example that you provided, if two 200bp reads overlap on 100 bases but are from a region (e.g. a highly conserved, essential gene) that contains very few SNPs (below $p$ SNPs), these two reads are overlapping but do not contain enough evidence (having the same alleles over at least $p$ SNPs) to be called  'consistent' and thus may still be 'ambiguous'.
>
> Q2: Following the answer for Q1, we agree that our choice of the term  'non-overlapping' may be confusing and we could revise it to 'ambiguous' to mean that there is not enough evidence to support that these two read should belong to the same haplotype ('consisten') or should belong to the different haplotypes ('conflict').
>
> Therefore, in the example for `two overlapping reads with e.g. $q-1$ differing SNPs'. These two reads are 'ambiguou' (not 'non-overlapping') because there is no enough evidence to support the 'Consistent' or 'Conflicting' relationship between them.
>
> Q3: Thanks for providing this excellent example! The constraints that ``any two conflicting reads have two different colors and any two consistent reads have the same color" (line 185 in the original submission) are mainly used to explain the graph coloring problem and emphasise its difference with respect to the classic graph coloring problem (line 187-191 in the original submission). While the first conflict constraint 'any two conflicting reads have two different colors' makes sense, we agree with the reviewer that the second consistent constraint 'any two consistent reads have the same color' seems too strong as illustrated by the example constructed by the reviewer. In fact, in NeurHap, these two constraints have been incorporated into Eq.(3) with different importance/weights (1 vs $\lambda \in (0, 0.1)$ in all our experiments, refer to Experimental Setup), i.e., the conflict constraints are much stronger than consistent constraints.
>
> Therefore, in the example given above, when 1 is consistent with 2 & 3, but 2 & 3 are in conflict, NeurHap tends to assign different colors to 2 & 3 (satisfying the conflict constraint) and to assign 1 a color same as 2 or 3. This color assignment by NeurHap aligns with the example as there is clear that 2 & 3 should come from different haplotypes (i.e., the 2nd and 3rd haplotypes, AACCTT and AACCGG, respectively in the above example) while there is not enough evidence that 1 should comes from, equally likely from the second or the third haplotype).
>
> Q4: The neural network-based search/learning model is inspired by RUN-CSP. Compared to RUN-CSP, our method is novel with respect to the following. First, our model is simpler than RUN-CSP.
> We use simple and effective MLP to capture the topology information instead of a RNN (LSTM or GRU)-based model used in RUN-CSP as the learning module. Second, we take into consideration both conflict and consistent constraints in NeurHap-search, which leads to better performance compared with modeling only conflict constraints in RUN-CSP.
>
> Refer to the following table for a detailed comparison between RUN-CSP and NeurHap-search on Real-Potato and 5-strain HIV. We previously show the MEC scores of NeurHap and RUN-CSP in the supplementary material A.4, and we also add the running time here.
>
> |Data|Model | | #1 | #2 | #3|  #4 | #5 | #6 | #7 | #8 | #9 | #10 |
> | :--------  | :-----  | :----:  | :--------  | :-----  | :----:  |:----:  |:-----  | :----:  |:----:  |:-----  | :----:  |:----:  |
> | Real-Potato | RUN-CSP | MEC |186 |358 |107 | 1 | 185 | 890 | 553 | 492 | 647 | 767 |
> | | | Time | 36s | 59s | 63s | 53s | 49s | 61s | 47s | 49s | 67s | 58s |
> | | NeurHap | MEC | 183 | 363 | 93 | 1 | 172 | 889 | 532 | 399 | 595 | 683 |
> | | -search| Time | 28s | 33s | 37s | 39s | 36s | 39s | 36s | 34s | 39s | 40s |
> | 5-strain HIV | RUN-CSP | MEC | 2226 | 2375 | 2175 | 2408 | 2192 | 2748 | 2449 | 2614 | 2312 | 2567 |
> | | | Time | 408s | 393s | 361s | 365s | 382s | 416s | 368s | 370s | 366s | 369s |
> | | NeurHap | MEC | 1436 | 1525 | 1536 | 1346 | 1479 | 1448 | 1460 | 1430 | 1380 | 1495 |
> | | -search| Time | 72s | 69s | 71s | 70s | 73s | 74s | 71s | 72s | 69s | 74s |
>
> Besides, this paper has two contributions. First, we provide a unique formulation of the haplotype phasing problem as a graph coloring problem by constructing a read-overlap graph. Second, we also propose a local refinement strategy to adjust colors and further optimize the MEC scores.
>
> Q5: We will revise the toy example to show the challenges of the problem.

---

> > ### Comment · Reviewer_uUcF · 2022-08-08
> > **response to rebuttal**
> >
> > Thank you for providing clarifications / edits on the setup and definitions used in the model - these resolve my concerns. I am raising my score from 3 to 4 to reflect it.
> >
> > Regarding novelty, while the method is an improvement on the RUN-CSP [Toenshoff et al, Front. Artif. Intell 2021], I believe the new elements in the method (MLP instead of RNN, two types of edges instead of one; formulating phasing as graph coloring is in itself not novel, see e.g. HapColor, IEEE BIBM 2015) are more relevant to a venue like ISMB than to the NeurIPS community.

---

> > > ### Author Response · Authors · 2022-08-09
> > > **Author Responses to Reviewer uUcF**
> > >
> > > Thanks for the comments.
> > >
> > > We would like to clarify that there are two differences between HapColor and NeurHap. Firstly, HapColor constructs a Weighted Fragment Conflict Graph (WFCG), which differs from our proposed read-overlap graph that consists of conflicting and consistent edges. Secondly, HapColor proposes a color merging method by iteratively finding two colors with minimal merging costs. NeurHap-search offers a graph neural network to learn vertex representations and color assignments, followed by NeurHap-refine, a local refinement strategy to adjust colors and optimize MEC scores.
> > >
> > > Our paper formulates the haplotype phasing problem as a graph coloring problem and develops an algorithm based on graph representation learning and combinatorial optimization. It is an exploration of artificial intelligence in the field of biology. We believe it is well aligned with the topic listed in NeurIPS Call for Papers this year: `Machine Learning for Sciences (e.g. biology, physics, health sciences, social sciences)' (https://neurips.cc/Conferences/2022/CallForPapers). Thank you again for helping us make the paper clearer and stronger.

---

### Official Review · Reviewer_Exv5 · 2022-07-12

**Rating:** 7
**Confidence:** 3
**Soundness:** 3 good
**Presentation:** 3 good
**Contribution:** 3 good

**Summary:**

The authors propose a new method for haplotype reconstruction in polyploid species and viral quasispecies. Their approach relies on graph message passing neural network.

**Questions:**

I see how Eqn. 3 is a relaxation of the constraints in Eqn. 2, but how exactly does the objective (the Hamming distance) come in? Could you provide a proof/derivation of Eqn. 3 from Eqn. 2?

Line 94 typo: ployploid -> polyploid
Line 287 typo: quasiqpecies -> quasispecies
Table 3: title says 3 viral quasispecies datasets, but 10 datasets are listed
Where are the colors from in Figure 3? (i.e. what’s the reference haplotype assignment?)

**Limitations:**

Although these are not problems unique to this method, I think the method’s inability to take into account statistical uncertainty, and inability to automatically discover the number of haplotypes (as may be necessary in studying viral quasispecies) are important limitations that should be discussed and addressed going forward.

**Strengths And Weaknesses:**

The paper offers a clear and careful explanation of the problem, and an original solution. The performance gains over previous methods are substantial, suggesting the method could likely have significant impact on biological practice.

A weakness of the method is that the problem is not set up using an explicit generative model, with a clear description of the estimand and noise. This makes it difficult to understand the statistical properties of the estimation problem the authors are trying to solve. Instead, an objective is simply assumed (though I understand it is a standard objective in the field). Perhaps even more importantly, this also prevents connections to the broader ML/statistics literature, and in particular prevents application/comparison to well-studied methods for clustering vectors of discrete data, with entries missing at random (an alternative view of the haplotype reconstruction problem).

---

> ### Author Response · Authors · 2022-08-01
> **Author Responses to Reviewer Exv5**
>
> Thanks for the comments.
>
> Q1: While Eqn. 2 aims to minimize the sum of hamming distances between each read $R_j$ and the haplotype $H_i$ that $R_j$ is drawn from, Eqn. 3 aims to minimize the divergence between pairs of reads (as $P(v_i)$ and $P(v_j)$) that are drawn from the same haplotype and maximize the divergence between pairs of reads if they are drawn from different haplotypes. Moreover, the hamming distances in Eqn.2 have been used implicitly to derive $E_{\neq}$ and $E_{=}$ in Eqn.3, i.e., two reads are ' in conflict' if the Hamming distance between two reads is larger than $q$; are ' consistent' if the Hamming distance between them is 0; are 'ambiguous' otherwise. Note that we have replaced the term 'non-overlapping' by 'ambiguous' to avoid possible confusion as suggested by another reviewer.
>
> In an ideal case, if all pairs of conflict reads are assigned into different haplotypes (i.e., different colors) and all pairs of consistent reads are assigned into the same haplotypes (i.e., the same color), each cluster will only contain consistent reads and thus the MEC score in Eqn.2 will be minimized to be 0. In non-ideal cases such as Real-Potato Sample 1 datasets, the following table shows that the objective function to be minimized in Eqn.2 (i.e., MEC) correlates well with the objective function to be minimized in Eqn.3, which demonstrates the effectiveness of NeurHap for minimizing the MEC through optimizing Eqn.3.
>
> |Loss value of Eqn.3| 1.46 | 1.31 | 1.01 | 0.83 |  0.71 | 0.66|
> | :--------  | :-----  | :----:  | :--------  | :-----  | :----:  | :----:  |
> |MEC score in Eqn.2 |739 | 415|302 | 265 | 220 | 187 |
>
>
> Q2: We will fix these typos in our revision.
>
> In Table 3, we show the experimental results on three viral quasispecies datasets (5-strain HIV, 10-strain HCV, and 15-strain ZIKV). Ten samples are generated by randomly sampling from each of these three datasets. Thus, Table 3 contains tree datasets with ten small datasets or samples in each of the big datasets.
>
> In Figure 3, Grey edges are conflict edges and blue edges are consistent edges in the read-overlap graph.
> The colors in Figure 3 are derived from the clusters constructed by different models, i.e., each color indicates a cluster of reads that are inferred to come from a same haplotype. We will add more details to the caption.
>
>
> Q3: Thanks. We will add these limitations in our concluding discussion.

---

### Meta-Review · Area_Chair_mMff · 2022-08-26

**Recommendation:** Accept
**Confidence:** Less certain

**Metareview:**

This paper presents a method, NeurHap, that addresses the haplotype reconstruction problem using a combination of graph representation learning and combinatorial optimization. The reviewers engaged in a detailed discussion with the authors on the merits of the paper. Some reviews found weakness in the novelty, yet as a contribution to the problem domain, found it useful. The main concerns were around lack of clarity. One review found a lack of clarity about the estimand, noise and objective which limit connections to work in the broader machine learning community. There was also a lack of clarity in terminology around "consistent" and "ambiguous". Finally, there was a lack of clarity around the parameters used in the method for the data analysis. These concerns compounded to reduce enthusiasm for the paper. I most strongly encourage the authors to take the reviewers' comments into consideration and revise their paper to improve the clarity and consistency with existing literature in the field so that the contributions can be understood and appreciated by the wider machine learning community.


**Award:**

No

---

### Decision · Program_Chairs · 2022-09-14

Accept